# MoteS: Memory Optimization via Fine-grained Scheduling for DNN on Tiny Devices

## Abstract

There has been a growing trend in deploying deep neural networks (DNNs) on tiny devices. However, it is challenging to do so due to the contradiction of large execution memory requirement of many DNNs and stringent memory constraint of tiny devices. Some previous works incurs large latency overhead to save memory and cannot optimize networks with complex structures; some methods employ coarse-grained scheduling for complicated networks, leading to limited memory footprint reduction. This paper proposes MoteS that performs fine-grained scheduling via operator partitioning on arbitrary DNNs to dramatically reduce peak memory usage with little latency overhead. MoteS presents a graph representation named Axis Connecting Graph (ACG) to perform operator partition at graph-level efficiently. MoteS further proposes an algorithm that searches the partition and schedule guided by memory bottlenecks. We evaluate MoteS using various popular networks and show that MoteS achieves up to 80% of peak memory usage reduction compared to state-of-art works with nearly no latency overhead on tiny devices.

## 1 Introduction

There has been a growing trend in deploying deep neural networks (DNNs) on tiny devices such as micro-controller units (MCUs), facilitating ubiquitous application of AI in Internet of Things (IoT) area. The tiny device has simple storage architecture, including an SRAM of usually no more than 2 MB and an extensible read-only Flash of several MBs. For instance, the STM32F767ZI has a 512 KB SRAM and an extensible Flash with initial capacity of 2 MB. Generally, when deploying models on tiny devices, model weights are allocated to Flash with relatively sufficient capacity (Banbury et al., 2021), while the intermediate tensors during inference must be allocated to the capacity-limited SRAM. However, even networks tailored for resource-limited devices require large memory compared to the limited SRAM of tiny devices. Some networks like

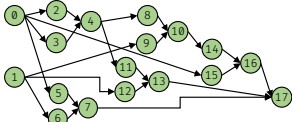

**(a)** A cell of a complicated network

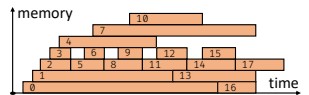

**(b)** Timeline and memory usage of (a)

Figure 1

MobileNetV2 (Sandler et al., 2019) / BERT-Tiny (Devlin et al., 2018) need large image resolution / long token sequence for accuracy. Some networks have complicated structure to enhance model expressiveness (Zoph et al., 2018; Real et al., 2019; Cui et al., 2019; Liu et al., 2019a; Xie et al., 2019; Devlin et al., 2018). Figure 1 (a) displays the structure of a cell in networks like NASNet (Zoph et al., 2018). These complex structures often result in intermediate tensors with long lifetimes during the inference process, as shown in Figure 1 (b). These long-lived tensors occupy a considerable amount of memory.

Previous work (Huang et al., 2020; Ahn et al., 2020; Wang et al., 2022) find that carefully choosing the execution order of operators in networks with complicated structures can effectively reduce memory usage. However, these memory optimization is limited because they consider only coarse-grained scheduling, that is, scheduling the network in an operator-by-operator manner. We find that exploring the scheduling space inside the operator can better reduce memory usage. By partitioning the operators into smaller ones and scheduling the finer-grained graph, more memory reduction can be achieved. For example, Figure 2 (a) shows that when we schedule the graph coarsely, we can optimize peak memory usage to at most 768. But if we enable more fine-grained scheduling, as

shown in Figure 2 (b), we can reduce peak memory usage to 384. This insight can be intuitively understood as follows: each output value of a sub-graph may depend only on a subset of input values. With good partition and scheduling, different blocks within the same original tensor can have non-overlapping lifetimes, and the memory of them can be shared. For example, the tensor of operator $v_1$ in Figure 2 (a) is partitioned into the tensors of operators $v_3$, $v_6$, $v_{11}$, $v_{14}$ in (b). Under the schedule of (b), these tensors' lifetime intervals are disjoint, so they can share the same memory with size $4 \times 32$ instead of the memory of the original tensor with size $8 \times 64$.

However, fine-grained scheduling makes operator shapes smaller and the amount of operators larger, which results in lower hardware utilization and more kernel invocations, incurring latency overhead. Therefore, memory optimization via fine-grained scheduling needs to be performed under certain latency constraint. But there are quite a lot of partition schemes for each operator in the network, which together form a large optimization space. How to search for a network partition scheme and corresponding schedule to minimize peak memory under the latency constraint is challenging. Meanwhile, for operators with overlapped sliding-windows such as convolution or pooling, partitioning the output operator

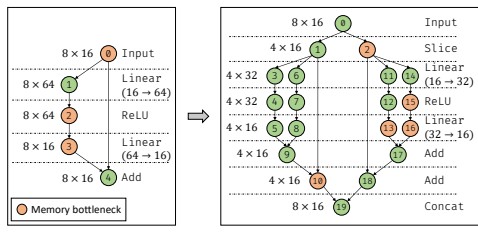

**(a)** Peak Memory: 768      **(b)** Peak Memory: 384

Figure 2: The graph & schedule & peak memory usage before and after fine-grained scheduling with the help of operator partition, where the numbers in nodes indicate execution order.

may cause overlapped input dependency, leading to a large amount of re-computation overhead (Lin et al., 2021; Stahl et al., 2023). How to deal with overlapped sliding-windows also poses challenges.

To address these challenges and enable efficient fine-grained scheduling, we propose a memory optimizer, called MOTES. We find that since peak memory usage is determined by the maximum memory usage during execution, many combinations of partition strategies for operators have the same effect. Therefore, it is inefficient to explore each partition strategy for each operator separately. We introduce a graph representation called Axis Connecting Graph (ACG) to consider partitioning operators at the graph level, significantly reducing the optimization space. With the help of ACG, we design an optimization algorithm to select sub-graphs based on the current memory bottleneck to further reduce the optimization space. In addition, to avoid the large computation overhead caused by overlapped sliding-windows, we can carefully manage and allocate memory for the input blocks that are dependent by multiple output blocks to eliminate computation overhead.

In summary, this paper makes the following contributions:

- We propose MOTES, a memory optimizer using fine-grained scheduling for DNN.
- We propose a graph representation named "Axis Connecting Graph" (ACG) and design an optimization algorithm based on ACG and memory bottleneck to partition and schedule network to reduce its peak memory usage.
- We evaluate MOTES extensively with popular vision and language models, e.g., MobileNetV2 (Sandler et al., 2019), MCUNetV2 (Lin et al., 2020), BERT-Tiny (Devlin et al., 2018), etc, and networks with complex structures, e.g., NASNet (Zoph et al., 2018).

Experiment results shows that MoteNeT can reduce up to $80\%$ peak memory usage compared to state-of-the-arts scheduling frameworks with less then $5\%$ latency overhead, enabling many powerful networks deployed on memory-limited tiny devices.

## 2 RELATED WORK

**Model Compression.** Several techniques have been developed to compress the parameters of a given model to deploy it on resource-constrained devices. Pruning removes redundant parameters (Liu et al., 2017; 2019b; Lin et al., 2017; He et al., 2017; 2018; Han et al., 2015b). Quantization reduces bit precisions for both parameters and activations. (Choi et al., 2018; Courbariaux et al., 2016; Han et al., 2015a; Rastegari et al., 2016; Wang et al., 2019; Zhou et al., 2016; Zhu et al., 2016). Tensor decomposition reduces redundant ranks of parameters (Gong et al., 2014; Lebedev et al., 2015; Kim et al., 2016). Neural Architecture Serach optimizes structures of networks or shapes of layers to reduce the number of parameters (Zoph et al., 2018; Real et al., 2019; Cui et al., 2019; Liu et al.,

2019a; Xie et al., 2019; Sandler et al., 2019; Cai et al., 2019; Tan et al., 2019; Cai et al., 2020; Lin et al., 2020).

Table 1: Comparison with prior-art methods of network inference on tiny devices

| Method | Network Structure | Memory Reduction | Latency Overhead | Techniques |
|---|---|---|---|---|
| CMSIS-NN (Lai et al., 2018) CMix-NN (Capotondi et al., 2020) | Arbitrary | No | No | Library Support |
| TFLite-Micro (David et al., 2021) Micro-TVM (Chen et al., 2018) | Arbitrary | No | No | Runtime Support |
| TinyEngine (Lin et al., 2020; 2021) FDT (Stahl et al., 2023) | Simple | High | Medium | Patch-based Inference; Depth-wise Tiling |
| Serenity (Ahn et al., 2020) HMCOS (Wang et al., 2022) | Arbitrary | Medium | No | Coarse-grained Scheduling |
| **MOTES (Ours)** | **Arbitrary** | **High** | **Low** | **Graph-level Analysis, Fine-grained Scheduling** |

**Network Inference on Tiny Devices.** Deploying neural networks on tiny devices has been a hot topic in recent years. Table 1 shows the comparison of MOTES related works. CMSIS-NN (Lai et al., 2018) and CMix-NN (Capotondi et al., 2020) provide library support for accelerating neural network operators on MCUs based on ARM Cortex-M chips. TFLite-Micro (David et al., 2021) and Micro-TVM (Chen et al., 2018) provide framework support for model inference on tiny devices. TinyEngine (Lin et al., 2020; 2021) reduces memory usage via patch-based inference for the initial layers of linear CNNs, and FDT (Stahl et al., 2023) reduces memory usage via depth-wise tiling along channel dimension of CNNs; such techniques can be transformed into a simple case of the fine-grained scheduling discussed in this paper. However, these methods will introduce large computation overhead for overlapped sliding-windows of convolutions, and they are hard to directly support more complicated networks like NASNet (Zoph et al., 2018), BERT-Tiny (Devlin et al., 2018) etc. For a complex network, there are many valid schedules, and different schedules can lead to different peak memory usage. Serenity (Ahn et al., 2020) uses a dynamic programming algorithm to find the schedule that minimizes peak memory usage. HMCOS (Wang et al., 2022) proposes an algorithm to cluster the network into hierarchical groups and performs group-wise scheduling. These methods perform coarse-grained scheduling, resulting limited memory reduction as illustrated in Figure 2.

## 3 PRELIMINARIES

**Computation Graph.** A DNN is often represented as a graph $G$. $V = \mathcal{V}(G)$ represents operators, where each operator has several input tensors and one output tensor (we might interchangeably use the terms node, operator, and tensor in this paper, since each node/operator corresponds to an output tensor). $E = \mathcal{E}(G) \subseteq V \times V$ represents the dependencies between operators, such as $(v_1, v_2) \in E$ indicates that operator $v_2$ depends on $v_1$ (usually, this means that the output tensor of $v_1$ is one of the input tensors of $v_2$). We use $\mathrm{pre}(v)$, $\mathrm{suc}(v)$, $\mathrm{anc}(v)$, $\mathrm{des}(v)$, $\mathrm{\#axes}(v)$, $v@i$, $|v@i|$, $\mathrm{size}(v)$ to represent the predecessors, successors, ancestors, descendants, number of axes (dimensions), the $i^{\mathrm{th}}$ axis of $v$, the length of $v@i$, and output tensor size of $v \in V$, respectively.

**Graph Schedule & Memory Usage & Memory Bottlenecks.** A topo-order $T = (v_1, v_2, ..., v_n)$ of $\mathcal{V}(G)$ is a schedule of graph $G$. Assuming that $i$ represents the time point when the $i^{th}$ operator is completed, we can get the lifetime of each operator $v_i$: $\mathrm{start}(i) = i - 1$, $\mathrm{end}(i) = \max_{v_j \in \mathrm{suc}(v_i)} j$. The set of tensors that are alive during the execution of operator $v_i$ is $\mathrm{alive}(i) = \{v_j \in T \mid \mathrm{start}(j) \leq i \leq \mathrm{end}(j)\}$. The memory usage of operator $v_i$ during execution is $M_i = \sum_{u \in \mathrm{alive}(i)} \mathrm{size}(u)$; and the **peak memory usage** during the execution of $G$ is $M_{peak} = \max_{1 \leq i \leq n} M_i$. The tensor that contributes to the peak memory usage is called **memory bottleneck**: $V_{mb} = \bigcup \{\mathrm{alive}(i) \mid M_i = M_{peak}\}$.

**Example.** In Figure 2 (c), during $v_3$'s execution there are three tensors alive: $v_0$, $v_2$, and $v_3$. Therefore, $M_3 = 8*16 + 8*64 + 8*16 = 768$, which is the peak memory usage of this graph under such schedule. Thus $v_0, v_2, v_3$ are memory bottlenecks. In Figure 2 (d), during $v_{16}$'s execution there are five tensors alive: $v_2, v_{10}, v_{13}, v_{15}$, and $v_{16}$. Therefore, $M_{16} = 4*16 + 4*16 + 4*16 + 4*32 + 4*16 = 384$, which is the peak memory usage of this graph under such schedule. Thus $v_2, v_{10}, v_{13}, v_{15}, v_{16}$ are memory bottlenecks.

## 4 METHODS

### 4.1 INSIGHTS OF MEMORY OPTIMIZATION VIA FINE-GRAINED SCHEDULING

We aim to optimize the peak memory usage, which refers to the maximum memory occupancy during execution. Our methods are based on two insights.

**Insight 1:** *It is inefficient to consider the operator partition separately for each operator*, as this may introduce much redundancy and hard to model the connections among operators. For instance, as shown in Figure 3 (b), splitting the two Linear operators separately in (a) has no impact on the peak memory usage. We need to consider the partition and scheduling of operators at the graph level.

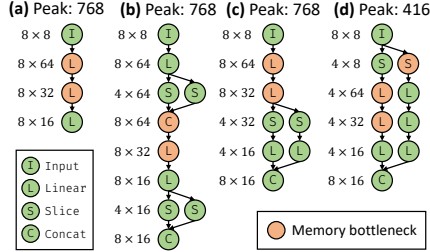

Figure 3: Different partition choices result in different peak memory usages.

**Insight 2:** *It is important to focus on memory bottlenecks*, like the tensors of size $8 \times 32$ and $8 \times 64$ in Figure 3 (a). The partition of non-bottleneck tensors like (c) doesn't impact the peak memory usage, while (d) splits bottlenecks of (a) and nearly halves the peak memory usage. In practice, memory bottlenecks are mostly concentrated inside certain cells (Wilken et al., 2000; Ahn et al., 2020). *Thus cells are appropriate sub-graphs for operator partitioning.*

### 4.2 WORKFLOW OVERVIEW

Figure 4 shows the workflow of MOTES. It accepts a model description like ONNX as input. The analyzer analyzes the model and builds an Axis Connecting Graph (ACG) based on the axis mappings of each operator and the dependency between operators. The optimizer then searches for feasible partition schemes and corresponding schedules based on ACG and memory-bottlenecks. It uses a profiler & simulator to measure the latency of operators with specific shapes and estimate the latency of the entire graph.

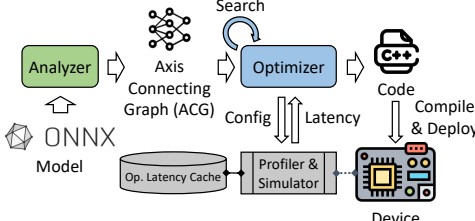

Figure 4: Workflow Overview of MOTES

Then, MOTES generates code based on the searched fine-grained graph and operator schedule. Finally, the generated code is compiled into a binary file for deployment on the target device.

### 4.3 AXIS CONNECTING GRAPH (ACG) FOR GRAPH-LEVEL OPERATOR PARTITION

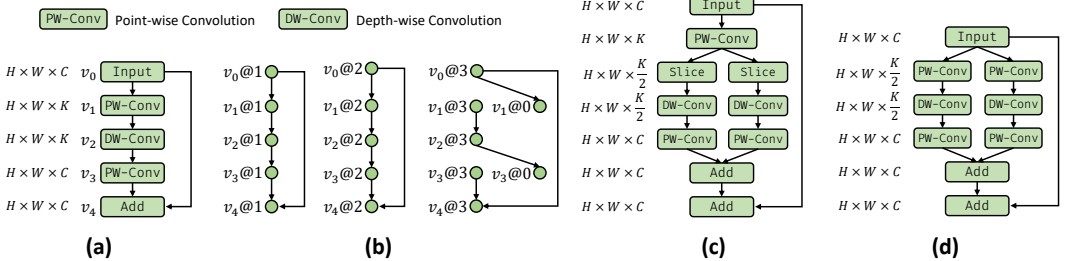

Figure 5: Example of Axis Connecting Graph (ACG) **(a)** Graph of an inverted-bottleneck in MobileNetV2 (Sandler et al., 2019). **(b)** ACG of graph in (a). **(c)** Partition sub-graph $\{v_1, v_2, v_3\}$ into two parts along ACG $\{v_1@3, v_2@3, v_3@0\}$. **(d)** Partition sub-graph $\{v_0, v_1, v_2, v_3\}$ into parts along ACG $\{v_1@3, v_2@3, v_3@0\}$.

Based on **Insight 1**, we need to consider operator partition at the graph level. When considering partition at the operator level, we need to get the axes (spatial-axis, reduce-axis) of an operator and partition along these axes. Likewise, to consider operator partition and scheduling at the graph level, we need a data structure to represent the "axes" of a subgraph. To this end, we propose Axis Connecting Graph (ACG) that represents the connection among the axes of different operators in a graph. The "axis" of a subgraph in the computation graph is a subgraph of the ACG. ACG represents

the inter-axis dependencies of operators in the form of a graph, which allows us to consider operator partition from graph-level, greatly reducing the optimization space that needs to be explored.

Given a graph $G$, we define its **Axis Connecting Graph (ACG)** as $A = \mathcal{A}(G)$, where

1. $\forall v \in \mathcal{V}(G), i = 1, 2, ..., \#\texttt{axes}(v) \Rightarrow v@i \in \mathcal{V}(A)$.

2. $\forall v \in \mathcal{V}(G) : v$ has a reduce-axis $\Rightarrow v@0 \in \mathcal{V}(A)$.

3. $\forall (u, v) \in \mathcal{E}(G), i, j \geq 1 : |u@i| = |v@j| \wedge u@i, v@j$ correspond to a spatial-axis $\Rightarrow (u@i, v@j) \in \mathcal{E}(A)$.

4. $\forall (u, v) \in \mathcal{E}(G), i \geq 1 : u@i$ corresponds to a reduce-axis of $v \Rightarrow (u@i, v@0) \in \mathcal{E}(A)$.

For instance, consider a `Linear` operator node $v$ with shape $N \times K$ and its input node $u$ with shape $N \times C$. The first axis of $u$ and $v$ corresponds to the same spatial-axis, and the second axis of $u$ corresponds to the reduce-axis of $v$, yielding $(u@1, v@1), (u@2, v@0) \in \mathcal{E}(A)$. Figure 5 (a) presents the graph of an inverted-bottleneck in MobileNetV2 (Sandler et al., 2019), and (b) shows sub-graphs of its ACG, where there are four connected sub-graphs, corresponding to the height-axis of `Input`, width-axis of `Input`, channel-axis of `Input`, and channel-axis of `DW-Conv` respectively.

With the help of ACG, we can represent the operator partition at graph level. For a graph $G$ and its subgraph $S \subseteq G$, we define a **Partition Scheme** $p = (S, A, V_I, V_O, n)$, where $A$ is a connected sub-graph of $\mathcal{A}(S)$, $V_I \subseteq \mathcal{V}(S)$ represents the input nodes of $S$ (which can be explicitly split by `Slice`), $V_O \subseteq \mathcal{V}(S)$ represents the output nodes of $S$ (which can be explicitly merged by `Concat`), and $n$ is the number of partitions. The scheme satisfies the following conditions: **(1)** $\forall v \in V_O : \exists i \geq 0 :$ s.t. $v@i \in \mathcal{V}(A)$ (output nodes must be partitioned or reduced); **(2)** $\forall v \in \mathcal{V}(S) \setminus V_O : v@0 \notin \mathcal{V}(A)$ (non-output nodes must not be partitioned along reduce-axis).

Figure 6 illustrates how to partition a sub-graph based on partition scheme. There are two input and two output nodes, with $u_1$ being split while $u_2$ is not. $v_1$ is split along the spatial-axis, while $v_2$ is split along the reduce-axis. The subgraph is split into two parts, with $u_1$ being split into two parts using a `Slice` operator, and $u_2$ being reused in both parts. Parts of $v_1$ are merged using a `Concat` operator in the end, while parts of $v_2$ are accumulated using an `Add` operator. Figure 5 (c) (d) shows two examples of partitioning sub-graphs of (a) along sub-ACG of (b).

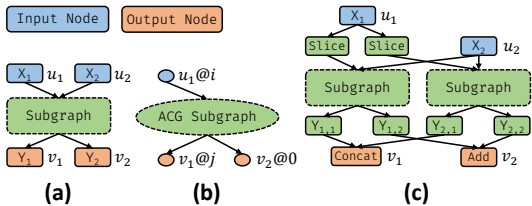

Figure 6: Illustration of partition. (a) Graph for partition. (b) ACG that partition along $(i, j > 0)$. (c) Result graph after partition.

## 4.4 HANDLING OVERLAPPED SLIDING-WINDOW

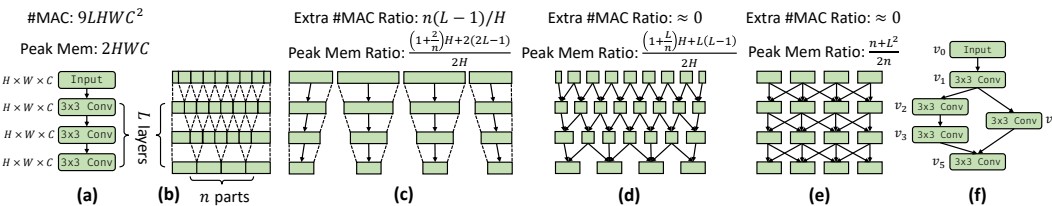

Figure 7: **(a)** $L$ consecutive $3 \times 3$ `Conv2d`. **(b)** Dependency between blocks if we split the output into $n$ parts. **(c) (d) (e)** Three solutions to split the subgraph. "Extra #MAC Ratio" means the ratio of extra #MAC to original one. "Peak memory ratio" means the ratio of current peak memory to original one. **(f)** Example of a subgraph where different paths have different numbers of `Conv2d`.

For an operator like convolution with kernel size greater than 1, a sliding-window reduction will be performed along the height/width axis, resulting enlarged receptive fields. If a sub-graph contains multiple such operators in a pathway, the receptive field enlargement accumulates, *leading to a significant amount of extra computation* (Lin et al., 2021). Figure 7 (a) shows $L$ consecutive $3 \times 3$ `Conv2d`, (b) shows the block-dependencies between layers when the sub-graph is split into $n$ parts starting from the output, and (c) shows the resulting partitioned sub-graph and its extra computation ratio as well as peak memory ratio. We can observe that to reduce memory usage, $n$ needs to be increased, resulting larger extra computation. To eliminate such overhead, we can infer the

blocks' bounds layer by layer based on their dependency, and split the overlapping parts among input blocks into separate blocks, as shown in Figure 7 (d). This method incurs no extra computation, but some intermediate tensors should reside in memory, so memory usage is slightly larger. However, *although such method introduces not extra computation, it leads to too many small operators, greatly influencing actual performance*, as illustrated in Table 3. Furthermore, the two methods above have a common drawback when dealing with many complicated CNNs, where different paths may have different numbers of operators with sliding-windows. As shown in Figure 7 (f), the receptive fields from $v_5$ to $v_1$ differ on the two paths due to the different numbers of Conv2d. *It is complicated to handle the sub-graph partition in such situation with the above methods.*

Therefore, we propose a compromised solution, which is to keep the partition granularity consistent across layers, as shown in Figure 7 (e). Before each block is executed, we use a BlockConcat kernel to merge the input blocks that it depends on (for some input blocks, only a portion of their values are consumed by output blocks, and we will only copy such portion). The overhead introduced by BlockConcat can be ignored in most cases as shown in Table 3. This method incurs more memory usage than the previous ones. For most cells of most networks, $L$ usually varies from 2 to 6 (He et al., 2016; Howard et al., 2017; Sandler et al., 2019; Zoph et al., 2018; Real et al., 2019; Cui et al., 2019; Liu et al., 2019a; Xie et al., 2019). Here, we take $L = 4$ and obtain the peak memory ratio of our solution is $\frac{n+16}{2n}$, which is larger than the method in Figure 7 (c) by $\frac{7}{n} - \frac{7}{H}$. As $n$ increases, i.e., when the overall memory usage is lower, the difference becomes smaller.

## 4.5 OPTIMIZATION ALGORITHM

---

**Algorithm 1:** Memory-bottleneck-aware Beam Search Algorithm

**input** : Graph: $G$. Latency constraint ratio: $\delta$. Beam width: $\beta$
**output** : Fine-grained graph after operator partition: $G'$. Schedule: $T'$

1   $\mathbf{C}_{rest} :=$ all cells of $G$;   $\mathbf{P} := \{\{\text{DummyPartitionScheme}\}\}$;
2   $G' := G$;   $T' := \text{Schedule}(G')$;   $L_\top := \text{Latency}(G') \times \delta$;
3   **def** LessThen($P_1, P_2$)**:**
4     **return if** $P_1.L \le L_\top \wedge P_2.L \le L_\top$ **then** $(P_1.M, P_1.L) < (P_2.M, P_2.L)$ **else** $(P_1.L, P_1.M) < (P_2.L, P_2.M)$;
5   **while** $true$ **do**
6     $V_{mb} := \text{MemBottlenecks}(G', T')$;   $\mathbf{C}_{mb} := \{C \in \mathbf{C}_{rest} \mid \mathcal{V}(C) \cap V_{mb} \ne \emptyset\}$;
7     **if** $\mathbf{C}_{mb} = \emptyset$ **then break**;
8     $C := \text{argmax}_{C \in \mathbf{C}_{mb}} \Sigma_{v \in (\mathcal{V}(C) \cap V_{mb})} \text{size}(v)$;   $\mathbf{C}_{rest} := \mathbf{C}_{rest} \setminus \{C\}$;
9     $\mathbf{A}_{mb} := \{A \in$ all valid connected sub-graphs of $\mathcal{A}(C) \mid \exists v \in V_{mb}, i \ge 1$ s.t. $v@i \in \mathcal{V}(A)\}$;
10    **for** $A \in \mathbf{A}_{mb}$ *sorted by depth* **do**
11      $\mathbf{P}' := \{P \cup \{p\} \mid P \in \mathbf{P}, p \in \text{GenSchemes}(A)\}$;
12      $\mathbf{P} := \text{MaxHeap}(\text{LessThen})$;
13      **for** $P \in \mathbf{P}'$ **do**
14       $G' := \text{Apply}(G, P)$;   $T' := \text{Schedule}(G')$;
15       $P.M := \text{PeakMem}(G', T')$;   $P.L := \text{Latency}(G')$;
16       $\mathbf{P}.\text{push}(P)$;   **if** $|\mathbf{P}| > \beta$ **then** $\mathbf{P}.\text{pop\_largest}()$;
17    $P := \mathbf{P}.\text{get\_smallest}()$; $G' := \text{Apply}(G, P)$; $T' = \text{Schedule}(G')$;
18 **return** $G', T'$;

---

With the aid of ACG, we design a beam search algorithm based on **Insight 2** to optimize peak memory under a given latency constraint, as shown in Algorithm 1. It iteratively partitions the graph (line 5-17) by identifying memory bottlenecks (line 6), selecting the cell containing bottlenecks (lines 8), and the connected sub-ACG containing bottlenecks within a cell (lines 9). $\mathbf{P}$ stores $\beta$ (hyper-parameter of beam search) optimization states, where each state $P$ contains several partition schemes $p$. Each sub-ACG $A$ can generate multiple partition schemes $p = (S, A, V_I, V_O, n)$ (with different partition numbers $n$), which are appended to current states to construct new ones (line 11). Then, top-$\beta$ smallest new states are kept for future iterations (line 13-16). We use a maximum heap to store new states and pop-out largest one when heap's capacity exceeds $\beta$ (line 16). Here we define a comparing function LessThen (line 3-4) for the heap (line 12), which compares peak memory first if latency meet the constraint otherwise compares latency first (line 4).

**Details.** GenSchemes generates different partition schemes $p = (S, A, V_I, V_O, n)$ for a given ACG $A$ by enumerating factors $n$ of the axis-length of $A$. Schedule is responsible for scheduling a fixed graph. We run this function frequently, and the graph it handles is complex after partitioning. Therefore, we choose the fastest reverse-post-order algorithm with linear complexity in terms of the network size; it can achieve near-optimal peak memory in most cases (Wang et al., 2022). Apply

applies the partition schemes on a given graph to produce a new fine-grained graph. `PeakMem` and `MemBottlenecks` are implemented using the formulas in Section 3. `Latency` measures the execution latency of the graph. It contains: (1) A profiler deployed on device to measure operator latency. (2) A simulator with an operator latency cache, storing the operator latency based on its type and shape; a simulator sums up the latency of all operators to get the total latency of the entire graph.

**Complexity.** Suppose the upper bounds of $|\mathbf{C}_{rest}|$ in line 1, $|\mathbf{A}_{mb}|$ in line 9, $|\texttt{GenSchemes(A)}|$ in line 11, and $|\mathcal{V}(G')|$ in line 14 are $N_C$, $N_A$, $N_n$, and $N_V$ respectively, then the asymptotic complexity of Algorithm 1 is $O(\beta N_C N_A N_n) \cdot O(N_V)$. Practically, $N_C$, i.e., the number of cells in a network, is no more than 20, and the actual iteration number of line 5-17 is generally less than 5; $N_A$ is generally less then 5; $N_n$ is no more then 10 in most cases due to the small lengths of axes of operators that can be deployed on tiny devices; $\beta$ is usually set as $4 \sim 8$. $O(N_V)$ is the complexity of `Schedule`, which is fast enough in practice. In our evaluation, all the optimization processes of MOTES conducted in Section 5 can be completed within 2 minutes (with $\beta = 8$).

## 5 EVALUATION

### 5.1 EXPERIMENT SETUP

We select the STM32H7A3ZI-Q MCU with ARM Cortex-M7 core as our test platform, which has 1.4 MB of SRAM but can be used to simulate lower memory constraints. We utilize CMSIS-NN (Lai et al., 2018) as the back-end operator library for MOTES and baselines and implement a kernel of `BlockConcat` proposed in Section 4.4 for MOTES. Besides, we adopt the in-placed depth-wise convolution kernel proposed in TinyEngine (Lin et al., 2020) instead of the one in CMSIS-NN for MOTES and baselines. We implement a tiny profiler deployed on the MCU to measure the operator execution latency given operator type and shape; the profiler communicates with the host via USB serial port.

We evaluate MOTES mainly with six popular networks, as shown in Table 2. Note that for NASNet-A (Zoph et al., 2018) and DARTS (Liu et al., 2019a), different datasets (input sizes) correspond to different architectures. For MCUNetV2 (Lin et al., 2021), we use the released model MCUNetV2-SE-Large. These networks have either large activation sizes or complex topologies, suffering from high peak memory during inference although they are tailored for resource-limited devices.

Table 2: Networks for Evaluation

| Network Name | Input Size |
| --- | --- |
| NASNet-A (CIFAR-10) (Zoph et al., 2018) | $32 \times 32 \times 3$ |
| DARTS (CIFAR-10) (Liu et al., 2019a) | $32 \times 32 \times 3$ |
| NASNet-A (ImageNet) (Zoph et al., 2018) | $224 \times 224 \times 3$ |
| DARTS (ImageNet) (Liu et al., 2019a) | $224 \times 224 \times 3$ |
| FPNAS (Cui et al., 2019) | $224 \times 224 \times 3$ |
| MobileNetV2 (Sandler et al., 2019) | $224 \times 224 \times 3$ |
| MCUNetV2 (Lin et al., 2021) | $224 \times 224 \times 3$ |
| BERT-Tiny (Devlin et al., 2018) | $512 \times 128$ |

We compare MOTES with two state-of-the-art open-sourced works of memory optimization for network inference on tiny devices: TinyEngine (Lin et al., 2020; 2021) and HMCOS (Wang et al., 2022). TinyEngine proposes patch-based inference to extremely optimize memory usage of simple structured networks like MobileNetV2 (Sandler et al., 2019). HMCOS is the state-of-the-art memory-aware graph scheduler for networks with complicated structures.

### 5.2 PEAK MEMORY USAGE REDUCTION

We evaluate and compare the peak memory usage optimized by TinyEngine, HMCOS, and MOTES on the benchmark networks. We run MOTES with latency overhead constraint $\delta = 1.05$ in Algorithm 1, that is, the execution latency overhead introduced by fine-grained scheduling should be no more than $5\%$. To demonstrate the effect of MOTES, we set several memory constraints (256KB, 320KB, 512KB, 1MB, and 2MB), which are common for tiny devices.

**Results.** Figure 8 (a) shows the evaluation results (with `int8` as quantization precision). For all networks, MOTES can achieve smaller peak memory usage than the baselines. MOTES achieves average $46\%$ (up to $72\%$) peak memory usage reduction compared to TinyEngine, and average $58\%$ (up to $87\%$) peak memory usage reduction compared to HMCOS. We also observe that MOTES can meet the 320KB memory limit for all the networks, while TinyEngine can satisfy such constraint for only MobileNetV2 and MCUNetV2, and HMCOS cannot satisfy such limit for any test networks.

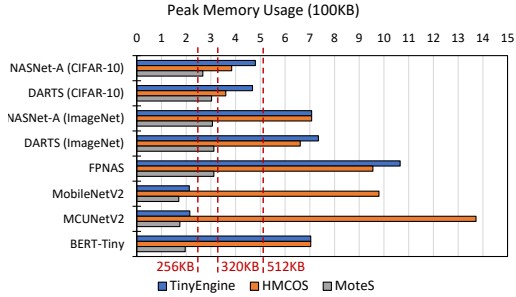

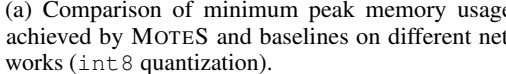

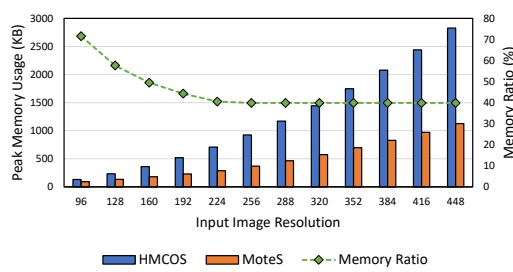

(a) Comparison of minimum peak memory usage achieved by MOTES and baselines on different networks (`int8` quantization).

(b) Peak memory (`int8`) of HMCOS and MOTES with different input image resolution for NASNet-A (ImageNet). "Memory ratio" refers to the ratio of memory usage between MOTES and HMCOS.

Figure 8

Furthermore, MOTES can meet the 256KB memory limit for 3 networks, while TinyEngine can satisfy it for only 2 networks. If we change the quantization precision from 8-bit to 16-bit or 32-bit, MOTES can easily meet memory constraints of 1MB or 2MB respectively on all benchmark networks, while TinyEngine / HMCOS can only satisfy 1MB (2MB) constraint on 4 / 2 networks for 16-bit (32-bit) quantization precision. We can observe that for NASNet, DARTS and FPNAS, HMCOS performs better than TinyEngine, since TinyEngine is hard to optimize networks with complicated structures. But for MobileNetV2 and MCUNetV2, TinyEngine is much better than HMCOS, since HMCOS cannot exploit intra-operator scheduling space, while TinyEngine can partially do it via patch-based inference. The reason why MOTES can achieve significant peak memory usage reduction is that it performs fine-grained scheduling via graph-level tensor partition, exploiting much more scheduling space inside each tensor compared with TinyEngine and HMCOS.

**Memory Usages with Different Input Sizes.** Note that for NASNet and DARTS with dataset CIFAR-10, memory footprint reduction of MOTES compared to HMCOS is at most $30\%$, while for dataset ImageNet, such reduction is at least $50\%$. Such variation is mainly because MOTES can exploit more memory reduction opportunities via fine-grained scheduling on networks with larger shapes. To further demonstrate the impact of input sizes, we compare MOTES with HMCOS under different input image resolutions. Figure 8 (b) shows the peak memory usage optimized by MOTES and HMCOS, as well as the ratio between the two, when changing the input image resolution of NASNet-A (ImageNet). As the input image resolution increases, the benefits of MOTES over HMCOS increases and saturates after the resolution gets larger than 224. This is consistent with the formula for peak memory ratio in Figure 7 (e), $\frac{n+L^2}{2n}$, as the number of partitions $n$ increases with higher input image resolutions, causing the value of $\frac{n+L^2}{2n}$ to decrease, but at a slower rate.

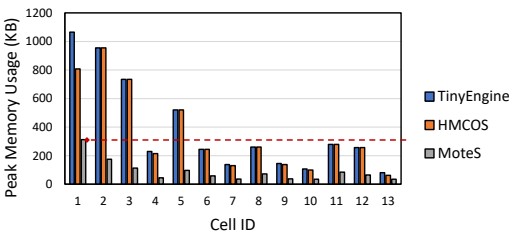

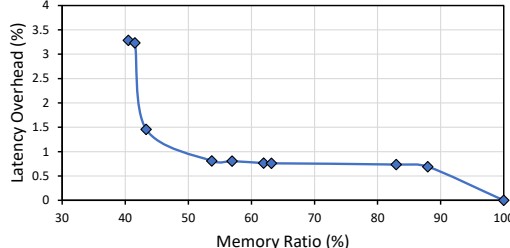

(a) Comparison of minimum peak memory usage achieved by MOTES and baselines on different cells of FPNAS network (`int8` quantization).

(b) Peak memory and latency overhead curve of NASNet-A (ImageNet). "Memory ratio" refers to the ratio of memory usage between MOTES and HMCOS.

Figure 9

**Memory Usage Breakdown of FPNAS.** Among all the test cases, MOTES can achieve the largest relative memory footprint reduction compared to baselines on FPNAS. This is mainly because they adopt inverted-bottleneck structure, which will greatly expand channel size temporarily. MOTES can exploit more memory optimization opportunities by partitioning some tensors along channel axis. Besides, FPNAS (containing `Slice`, `Concat`, and `Add` in each cell) is much more complicated than MobileNetV2 and MCUNetV2, so TinyEngine is hard to optimize it. We breakdown the cell-wise

peak memory on FPNAS. As shown in Figure 9 (a), MOTES can greatly reduce memory footprint on each cell. But the absolute memory reduction compared to baselines is smaller in deeper cells. After we perform tensor partition on cell 1, the major memory bottlenecks become cells 2, 3, and 5. In practice, our algorithm will only partition operators of the memory bottleneck cells to avoid unnecessary latency overhead.

### 5.3 LATENCY OVERHEAD EVALUATION

Table 3: Performance of partitioning $3 \times 3$ `Conv2d` along the height-axis and width-axis. The operator has a shape of `height=32,width=32,in-channel=8,out-channel=8.`

| Height & Width Partition Factor | 16 | 8 | 4 | 2 | 1 |
|---|---|---|---|---|---|
| Latency before Partition (us) | 6394 | | | | |
| Latency after Partition (us) | 6416 | 6448 | 6528 | 6656 | 9216 |
| Total Latency Overhead | 0.34% | 0.84% | 2.09% | 4.09% | 44.13% |
| `BlockConcat` Latency (us) | 28 | 48 | 108 | 281 | 819 |
| Latency Ratio of `BlockConcat` | 0.44% | 0.74% | 1.66% | 4.23% | 8.89% |

**Full Network Latency Overhead.** In this section, we analyze the latency overhead introduced by MOTES. We set the latency overhead constraint $\delta$ in Algorithm 1 as $1.05, 1.02, 1.01$ ($5\%, 2\%, 1\%$ latency overhead) and collect some sample points of memory ratio (the ratio between peak memory optimized by MOTES and HMCOS) and latency overhead of MOTES when optimizing NASNet-A (ImageNet). As shown in Figure 9 (b), to optimize memory ratio to $60\%$, the latency overhead is less than $1\%$, which is negligible. However, to further optimize it, the graph should be more fine-grained, resulting in more kernel invocations and lower hardware utilization due to smaller operators. Therefore, the latency overhead increases rapidly when the memory ratio is reduced.

**Single Operator Latency Overhead.** We further conduct an experiment to evaluate the performance degradation caused by partitioning a `Conv2d` operator along both height-axis and width-axis. The results are shown in Table 3. We can observe that as the factor decreases, the total latency increases faster and faster. For instance, reducing the factor from 16 to 8 only incurs less than $0.5\%$ extra latency overhead, while reducing it from 2 to 1 leads to $40\%$ extra latency overhead. Table 3 also shows that the latency of all `BlockConcat` kernels, relative to the total latency, is typically less than $2\%$. However, for extremely small partition factor, such ratio approaches to $10\%$ because of too many kernel invocations.

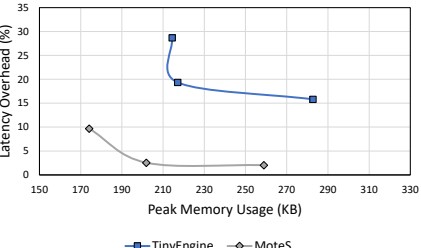

Figure 10: Peak memory usage and latency overhead curves of MCUNetV2-SE-Large optimized by TinyEngine and MoteS.

**Comparison to Latency Overhead of TinyEngine.** Besides, we present a comparison between MOTES and TinyEngine in terms of peak memory usage and latency overhead in MCUNetV2-SE-Large. For TinyEngine, we obtain three optimization results with patch sizes of 2, 4, and 7. For MOTES, we set the latency overhead constraint $\delta = 1.1$ and show three optimization states during search. Figure 10 shows the results. We can observe that both TinyEngine and MOTES can easily reduce peak memory from the original 1470KB to within 320KB. However, TinyEngine's latency overhead is much larger than MOTES due to the significant computation overhead caused by the patch-based inference, as shown in Figure 7 (c). In contrast, our approach can achieve comparable memory optimization with little additional computation overhead.

## 6 CONCLUSION

Memory optimization is critical to deploy DNN on tiny devices. This paper proposes MOTES which performs fine-grained scheduling for DNN to optimize peak memory. MOTES builds Axis Connecting Graph to represent graph-level axis partition, and search partition and schedule based on memory-bottlenecks. Our evaluation results show that MOTES can reduce up to $80\%$ peak memory usage of DNN compared to state-of-the-art works with nearly no latency overhead.

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

# A  APPENDIX

You may include other additional sections here.

