# OpenReview forum: "MoteS: Memory Optimization via Fine-grained Scheduling for DNNs on Tiny Devices"
_ICLR.cc/2024/Conference — ICLR 2024 Conference Withdrawn Submission_

### Official Review · Reviewer_uvi7 · 2023-10-30

**Soundness:** 3 good
**Presentation:** 3 good
**Contribution:** 2 fair
**Rating:** 6
**Confidence:** 4

**Summary:**

The paper addresses the challenge of deploying deep neural networks (DNNs) on small devices due to memory constraints. To overcome the limitations of latency and network complexity, this paper proposes MOTES to perform fine-grained scheduling by partitioning operators in DNNs. MOTES reduces peak memory usage with minimal latency overhead. MOTES utilizes the Axis Connecting Graph (ACG) as a graph representation for efficient operator partitioning and employs an algorithm guided by memory bottlenecks for partition search and scheduling. The evaluation results demonstrate that MOTES achieves up to 80% reduction in peak memory usage compared to existing methods, with negligible latency overhead.

**Strengths:**

The performance of MOTES is evaluated on popular neural networks, and the results demonstrate that it achieves up to an 80% reduction in peak memory usage compared to state-of-the-art methods, while imposing nearly no latency overhead on tiny devices. This makes MOTES a promising solution for optimizing memory when deploying DNNs on small devices.

**Weaknesses:**

(1) The effectiveness of the proposed model compression and learning strategy is closely tied to the processing capabilities of the underlying hardware. It is important to consider that not all commodity processing chips can support specific int8 operations or efficiently handle them. The true acceleration of sparse computing heavily relies on specific chips. It would be interesting to investigate whether the proposed methods can be generalized to other commodity devices, such as the NVIDIA Jetson or mobile phone neural chips, rather than being limited to the specific STM series.

(2) The experiments primarily focus on simple image classification with a limited number of training epochs, it would be valuable to assess the performance of the proposed methods on segmentation and detection tasks as well.

(3) Timely handling of the learning procedure is crucial. It is suggested to present the performance of training/inference speed or time cost when applying the proposed methods to downstream tasks. This would provide a more comprehensive understanding of the practical implications and efficiency of the approach.

**Questions:**

Please see the weakness part.

---

### Official Review · Reviewer_VBzs · 2023-10-31

**Soundness:** 2 fair
**Presentation:** 1 poor
**Contribution:** 2 fair
**Rating:** 3
**Confidence:** 3

**Summary:**

This paper proposes MOTES, a method for optimizing the peak memory usage of DNNs on tiny devices. It addresses the limitations of prior work by performing fine-grained scheduling through operator partitioning of networks. MOTES presents a graph-based representation called Axis Connecting Graph (ACG) to efficiently partition models at the graph level. Additionally, MOTES introduces an algorithm to search for partitions and schedules that are guided by identifying memory bottlenecks. The algorithm aims to minimize peak memory usage during deployment. Evaluation on popular neural networks demonstrates that MOTES is able to achieve dramatic reductions in peak memory usage of up to 80% compared to state-of-the-art techniques. Besides, this significant memory optimization is realized with nearly no increase in latency overhead for the tiny device setting.

**Strengths:**

1. The paper tries to addreess a meaningful problem in tiny-device DNN inference.

2. The approach of dividing operators along axes and finely scheduling them in the device inference scenarios demonstrates intriguing potential.

3. The evaluation results showcase promising outcomes, indicating the effectiveness and potential of the proposed method in device dnn inference.

**Weaknesses:**

1. The writing and presentation of this paper have huge room for improvement. Issues like incorrect figure references, grammatical errors, and lack of clarity in explaining key insights/definitions suggest the authors need to refine their work.

2. Operator/model partitioning during inference is not a novel idea. A referenced paper ((https://ieeexplore.ieee.org/document/9546452)) demonstrates such approach has been explored before in cloud/edge DNN settings.

3. Details of how operators actually execute during scheduling using the ACG are not fully explained.

**Questions:**

1. The paper does not clearly explain the execution flow within the generated ACG, which contain multiple branches due to operator partitioning along different axes. The execution sequence likely impacts performance, but this relationship is not investigated. More details on how operators are scheduled within an ACG would strengthen the evaluation of runtime impacts.

2. The authors could breifly illustrate how other baselines achieve the memory reduction.

3. Which compiler is used in the workflow?

4. In the related work part, the authors had better add content about model partition.

5.  (A minor point) In the introduction, it is mentioned that a longer token sequence can potentially improve accuracy. Unlike image resolution, which could be adjusted flexibly in inference, the length of sequence tokens of a request in natural language processing is typically fixed and cannot be changed unless it exceeds the maximum length allowed by the model. So It seems that the length of tokens does not have a direct relationship with the accuracy of the model. The authors had better check such expression.

---

### Official Review · Reviewer_MrR9 · 2023-11-01

**Soundness:** 1 poor
**Presentation:** 2 fair
**Contribution:** 2 fair
**Rating:** 3
**Confidence:** 5

**Summary:**

Deep Neural Networks (DNNs) deployed on edge devices are subjected to severe resource constraints, in particular, memory storage. This paper introduces a strategy to reduce peak memory usage in DNN inference by looking at the associated computation graph from an operator perspective and partitioning  it to minimize the peak memory usage of the partitioned operator subgraphs. The process of partitioning is executed on a so called "Axis Connecting Graph" (ACG) representation which is nothing but the original compuation graph viewed along different computation axes (e.g. horizontal/vertical splits, reduction e.g.,. result of a convolution etc.). They evaluate their method on several small DNNs from the literature and measure latency of their partitioning approach. From a high-level, partitioning of tensors seems to be the equivalent of classical {\it tiling}. Their partitioning algorithm is a heuristic that consists of trying several of the combinatorialy large partition space and choosing the best through a maintained heap structure. The measured performance metric is {\bf latency} although the partitioning motivation is actually peak {\it tensor memory storage size}.

**Strengths:**

The motivation behind the paper is sound in so far as the overall goal of scheduling edge DNN inference to minimize latency.

Looking at optimizations from an operator perspective may be an interesting future direction

The idea of task partitioning is a respectable direction of contemporary research in DNN acceleration.

**Weaknesses:**

(1) There seems to be a fundamental discrepancy between the parameters of the ACG graph (which is partitioned on the basis of peak memory usage) and the algorithm 1 objectives which is latency. There is a complex and unknown correlation between latency and peak memory usage, latency is affected by many other independent factors. Optimizing for peak memory usage is not the same as optimizing for latency. Thus many existing accelerators work on optimizing memory communication overhead i.e., memory traffic which has a direct correlation with latency. The approach followed in this paper towards optimizing latency through minimizing peak memory storage ignores many other factors.
(i) Reducing peak storage by increasing the number of partitions could increase memory traffic and therefore latency.
(ii) Other works look at local optimization criteria when considering optimal tile sizes (partitions) such as cache constraints, increased hit ratios etc. but none of that is considered in this paper.
(iii) The authors estimate latency of partitions through a run-time simulator which also adds overhead and could be incorrect.
(iv) What about the properties of the tensor, Memory usage is assumed the same whether it is a dense tensor or sparse tensor. Again existing works tile and do run time resource allocations based on sparsity.


(2) The definition of an Axis Connecting Graph is rather vague, it consists of a set of sufficient conditions. It is not clear how this definition actually allows you to get efficient partitions. It sems to be an exercise in definitions without an application. What, if any, is the rule for generating successively effective partitions using this definition of ACG?


(3) The paper seems to have been written hastily without proof reading, there are several critical omissions. For example,
(i) · In Section 3, the authors reference Figure 2 (c) and Figure 2 (d) when introducing an example, but these figures are missing in the paper. This example is very crucial to understanding their concept of peak memory usage and memory bottlenecks but since it is missing one has to guess.
(ii) The definition of memory bottleneck in Sec. 3 (bottom of page 3) is ambiguous and possibly incorrect. They seem to say that $start(i)$ is when $i-1$ is completed which can be assumed to be $end(i-1)$. However $end(i) = max_{j \in succesors(i)} j$, which is a recursive definition, i.e., i ends when its last successor ends which itself ends when its last successor ends and so on.
Such inconsistent and badly presented notation makes it hard to make sense of the rest of the paper. This needs to be proof read and cleaned up.

(4) Authors state that existing methods propose only coarse grained scheduling, however many such methods are based on solid optimization criteria that exploit algorithmic substructures for reducing the optimization search space, not just heuristics, e.g,, dynamic programming [Ahn2020].

(5) Authors state that techniques like depth tiling [Stahl 2023] are just special cases of their approach, which is technically true since their heuristic can generate all possible partitions through parameter $n$. Unfortunately this partitioning algorithm is brute force exhaustive, there is no exploitation of substructure to drive the algorithm towards an optimal partitioning solution. The authors make no claim as to whether they can even find the optimal.

(6) Note that every set of contiguous elements of a tensor is a trivial partition, thus the total number of partitions can be combinatorially explosive, it is not clear how they can reduce the search space in an algorithmically efficient manner to find the optimal.

**Questions:**

. What is the stopping criteria of Alg. 1?

· In Section 3, the authors reference Figure 2 (c) and Figure 2 (d) when introducing an example, but these figures are missing in the paper.

· In Section 4.2, the meaning of "reduce-axis" is unclear. The term needs to be explained or defined to enhance the paper's clarity.

· The computational complexity of the MemBottlenecks() function in Algorithm 1 is not well-defined. Finding new memory bottleneck after each iteration appears to be a non-trivial task and its computation complexity should be analyzed.

· The paper lacks consideration of the number of iterations when analyzing the complexity of Algorithm 1. What is the exact stopping criteria. You state that the number of iterations is around 5 but why?

· In Figure 7, it's unclear how the peak memory ratio is related to the kernel size.

---

### Official Review · Reviewer_kLuY · 2023-11-02

**Soundness:** 3 good
**Presentation:** 3 good
**Contribution:** 3 good
**Rating:** 6
**Confidence:** 3

**Summary:**

The paper is in the area of memory optimization for DNNs on small devices. It proposes MOTES that performs fine grained scheduling via operator partitioning on arbitrary DNNs to reduce peak memory usage with minimal latency overhead. Further it presents a graph representation ACG to perform graph level operator partition efficiently. It also presents an algorithm to search partitions and schedules based on memory bottlenecks. The evaluations show an 80% decrease in peak memory usage compared to state-of-the-art works.MoteNET reduces upto 80% peak memory usage compared to state-of-the arts with less than 5% latency overhead
Evaluation is done and gains are shown on standard and popular vision and language models making the impact of the work broader

**Strengths:**

- MoteNET reduces upto 80% peak memory usage compared to state-of-the arts with less than 5% latency overhead
- Evaluation is done and gains are shown on standard and popular vision and language models making the impact of the work broader

**Weaknesses:**

- The paper does not highlight limitations of the framework/approach and areas for future work in a detailed section, which can benefit the larger research community.
- While the latency improves compared to other approaches, it is still non trivial, and future work can be explored to reduce it even more.

**Questions:**

N/A